# PINCH: Accelerating Distributed GNN Training through In-Kernel Operation Using eBPF

Jianchang Su*, Yifan Zhang*, Wei Zhang*

*University of Connecticut,

*jianchang.su@uconn.edu, yifan.4.zhang@uconn.edu, wei.13.zhang@uconn.edu*

*Abstract*—Graph Neural Networks (GNNs) have achieved state-of-the-art performance on various graph-related tasks, but training GNNs on large-scale graphs remains challenging due to the high communication overhead of neighborhood aggregation. Existing distributed GNN training frameworks suffer from inefficiencies in communication due to frequent data movement between user and kernel space and the use of generic communication primitives.

We introduce PINCH, a novel system designed to speed up distributed GNN training. It employs eBPF and kernel hooks (XDP and TC) to shift communication-heavy operations to the kernel space. PINCH uses three main techniques: (1) in-kernel neighborhood aggregation via eBPF and XDP to cut communication costs, (2) in-kernel broadcasting through eBPF and TC to minimize user-kernel transitions and network stack passes, and (3) caching and reusing aggregated embeddings with eBPF maps to reduce redundant data processing. These integrations aim to alleviate the communication bottleneck and accelerate overall training.

## I. INTRODUCTION

Graph Neural Networks (GNNs) have emerged as a powerful framework for learning representations of structured data, achieving state-of-the-art performance on tasks such as node classification, link prediction, and graph classification [17], [25]. GNNs learn node embeddings by iteratively aggregating and transforming the features of neighboring nodes, allowing them to capture both node features and graph structure in the learned representations.

However, training GNNs on large-scale graphs remains challenging due to the high computation and communication costs involved, particularly in the context of the aggregation and broadcasting steps. In the aggregation step, each node gathers the embeddings of its neighbors to update its own representation [18]. This requires intensive communication between nodes, especially in distributed settings where nodes are partitioned across multiple machines. Similarly, in the broadcasting step, updated node embeddings need to be disseminated to all machines containing their neighbors, leading to significant communication overhead [8].

Several frameworks have been proposed to scale GNN training on distributed systems [6], [9], [31], and these frameworks employ techniques such as graph partitioning, parallelized mini-batch training, and layer-wise model parallelism to accelerate GNN computation. However, they still suffer from high communication overheads due to the need to frequently exchange node embeddings and gradients between workers during the iterative message passing process.

Recent works have explored communication-efficient algorithms for distributed GNN training to address this bottleneck. PipeGCN [23] proposes a pipelined message passing scheme that interleaves computation and communication to hide communication latency. Cluster-GCN [3] introduces a subgraph sampling strategy to reduce the number of nodes involved in each training iteration. CAGNET [19] presents a family of algorithms based on 1D, 1.5D, and 2D partitioning of the adjacency matrix to reduce communication volume and latency in the context of aggregation and broadcasting operations.

Despite these advances, the communication costs of neighborhood aggregation in distributed GNN training remain a significant bottleneck, especially on large, dense graphs and deep architectures. Motivated by this, we propose PINCH, a novel system that leverages eBPF (extended Berkeley Packet Filter) and kernel hooks (XDP and TC) to accelerate distributed GNN training by offloading communication-intensive operations to the kernel space. Our key contributions include:

- We introduce an in-kernel aggregation and broadcasting system using eBPF, XDP, and TC to minimize communication costs and reduce redundant crossings between user and kernel spaces.
- We introduce caching and reuse of aggregated embeddings across iterations using eBPF maps to minimize communication and computation overhead.
- We integrate the above techniques into a unified system, PINCH. We believe it will demonstrate its effectiveness in accelerating distributed GNN training on large graphs as the future work.

## II. BACKGROUND AND MOTIVATION

### A. Graph Neural Networks

GNNs have emerged as a powerful tool for learning representations of graph-structured data. GNNs iteratively aggregate information from neighboring nodes to learn node embeddings that capture both structural and feature information [18]. In the message passing framework, a GNN layer can be formulated as:

$$\mathbf{h}_i^{(l)} = \phi\left(\mathbf{h}_i^{(l-1)}, \bigoplus_{j \in \mathcal{N}(i)} \psi(\mathbf{h}_i^{(l-1)}, \mathbf{h}_j^{(l-1)}, \mathbf{e}_{ij})\right) \quad (1)$$

where $\mathbf{h}_i^{(l)}$ is the embedding of node $i$ at layer $l$, $\mathcal{N}(i)$ is the neighborhood of node $i$, $\mathbf{e}_{ij}$ is the edge feature between

nodes $i$ and $j$, $\psi$ is the message function, $\bigoplus$ is the aggregation function (e.g., sum, mean, max), and $\phi$ is the update function.

While GNNs have achieved state-of-the-art performance on various graph learning tasks, training GNNs on large-scale graphs remains challenging due to the high computational and memory requirements. To scale GNN training, distributed computing is necessary. However, in a distributed setting where the graph is partitioned across multiple machines, the neighborhood aggregation step in GNNs requires fetching node embeddings from other machines, leading to intensive communication. Specifically, at each GNN layer, each machine needs to broadcast the updated node embeddings to all other machines and wait for the aggregated embeddings from its neighbors. This communication overhead can dominate the training time, especially for deep GNNs on large graphs.

*B. eBPF and Hooks*

**eBPF** (extended Berkeley Packet Filter) is a Linux kernel technology that allows safely executing custom code inside the kernel. eBPF programs are written in a restricted C-like language and are verified by the kernel before being loaded and attached to specific hook points. This ensures the safety and stability of the kernel while enabling high-performance packet processing and monitoring. eBPF has been widely used for networking, security, and performance profiling applications [11].

**XDP** (eXpress Data Path) is an eBPF-based packet processing framework that executes programs early in the kernel's networking stack, right after packet reception. This enables XDP achieve low latency and high throughput by minimizing packet copies and context switches. AF_XDP extends XDP by providing a user-space interface, enhancing efficiency through zero-copy operations from the network driver to user space [22]. Both are widely used for DDoS mitigation, load balancing, and pre-filtering, efficiently handling network traffic and allowing advanced packet manipulation. with AF_XDP facilitating direct user-space access to enhance performance further [5].

**TC** (Traffic Control) is another eBPF hook point that allows attaching eBPF programs to the kernel's traffic control subsystem [1]. Unlike XDP, which processes raw packets, TC hooks work with Linux `sk_buff` structures that contain parsed packet information. TC allows eBPF programs to classify, filter, and manipulate packets based on various criteria such as IP addresses, ports, or custom markers. TC is commonly used for implementing complex network policies, quality of service (QoS) mechanisms, and virtual networking functionalities.

*C. Motivation*

The message passing nature of GNNs introduces significant communication overhead in distributed training, especially due to the need for broadcasting node embeddings and aggregating neighbor information at each iteration. Broadcasting node embeddings to all machines is necessary to ensure each machine has the required information for aggregation, while the aggregation step itself involves fetching and combining embeddings from different machines. As the graph size and model depth grow, this communication overhead can quickly become the scalability bottleneck, taking up to 80% of the total training time [2], [2], [14], [19].

Existing distributed GNN training frameworks, such as DistDGL [28] and DistGNN [10], rely on the traditional socket-based communication stack, which incurs high latency and CPU overhead due to frequent user-kernel crossings and network stack traversals. They also follow the conventional paradigm of pulling data from remote machines, leading to redundant communication and blocking waits.

We observe that the broadcasting and aggregation patterns in GNN communication are well-suited for applying in-network optimizations by leveraging the programmable packet processing capabilities of eBPF and kernel hooks (XDP and TC). By offloading these operations to the kernel space and performing them in a more efficient way, we can significantly reduce the communication overhead and improve the scalability of distributed GNN training

## III. DESIGN

Based on the motivation (Cf. § II-C), we propose PINCH, a novel system that leverages eBPF, XDP, and TC to optimize the communication in distributed GNN training. The key idea is to offload the broadcasting and aggregation operations to the kernel space and perform them in a more efficient way by utilizing the packet processing capabilities of eBPF. In this section, we describe the design of PINCH, with the following design goals:

**G1** Reduce the communication overhead of distributed GNN training.

**G2** Scalable to large graphs and deep GNN models.

**G3** Robust to network failures and ensure the correctness of the training process.

To achieve these goals, we employ the following techniques:

**In-kernel Broadcasting and Aggregation:** We offload the broadcasting and aggregation to the kernel space using eBPF programs attached to XDP and TC hooks.

**Kernel-bypass with AF_XDP:** We use the AF_XDP socket to transfer packets directly between the user space and the XDP program, bypassing the kernel network stack.

**Caching and Reuse:** We cache the aggregated results in eBPF maps to avoid redundant communication and computation.

*A. Overview*

PINCH consists of three main components: **In-XDP Aggregation**, **In-TC Broadcasting**, and **Caching**. The In-XDP Aggregation component performs neighbor embedding aggregation directly in the kernel upon receiving the packets. The In-TC Broadcasting component constructs and sends broadcast packets containing node embeddings. The Caching and Reuse component stores the aggregated embeddings in eBPF maps and reuses them whenever possible. The overall architecture of PINCH is shown in Figure 1.

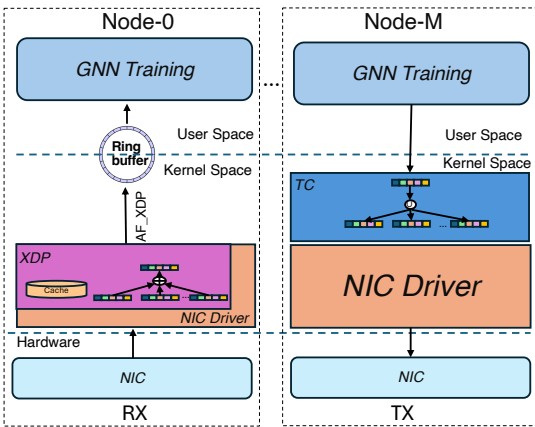

Fig. 1: Architecture of PINCH

### B. In-XDP Aggregation

The In-XDP Aggregation component is responsible for performing neighbor embedding aggregation directly in the kernel. It is implemented as an eBPF program attached to the XDP hook. Whenever a packet containing node embeddings arrives, the XDP program parses the packet, extracts the node ID and embedding, and performs the aggregation operation (e.g., sum, mean, or max) based on the specified GNN model. The aggregated results are stored in an eBPF map keyed by the node ID.

One challenge in implementing aggregation in XDP is that the eBPF virtual machine does not natively support floating-point operations, which are commonly used in GNN embeddings. To address this limitation, we adopt a scheme similar to prior works [8], [16], [26], where the workers multiply the floating-point embeddings by a large factor (e.g., $10^8$) and then round them to 32-bit integers before sending them to the aggregators. The XDP aggregation program performs integer addition operations on these quantized embeddings. When the workers receive the aggregated embeddings, they convert the 32-bit integers back to floating-point numbers by dividing them by the same factor and the number of workers.

Another challenge is the limited size of the eBPF stack and the restrictions on loops and function calls in the eBPF verifier. To handle these limitations, PINCH breaks down the aggregation logic into multiple smaller functions and uses eBPF tail calls to chain them together. The aggregation state is stored in eBPF maps, which can be accessed across tail calls and persist across packet processing.

To further reduce the overhead of transferring packets between the user space and the kernel space, we use the AF_XDP socket to enable kernel-bypass. With AF_XDP, the packets can be directly transferred between the user space and the XDP program, bypassing the kernel network stack. This can significantly reduce the latency and the CPU overhead of packet processing.

The aggregated embeddings are cached in an eBPF map. If the same set of nodes are aggregated again in a later iteration,the cached results can be directly reused as described in § III-D. The In-XDP Aggregation component also handles

deduplication as described in § III-E.

### C. In-TC Broadcasting

The In-TC Broadcasting component is responsible for constructing and sending broadcast packets containing node embeddings. It is implemented as an eBPF program attached to the TC hook.

Inspired by the message broadcasting mechanism in Electrode [30], we leverage the `bpf_clone_redirect()` helper function to efficiently clone and redirect packets for broadcasting. This function allows the eBPF program to create multiple copies of a packet and redirect them to different network interfaces or sockets without incurring additional user-kernel crossings or memory copies.

The broadcasting process works as follows. When the user-space GNN framework has finished updating the embeddings of a batch of nodes, it passes these embeddings to the TC program via an eBPF map. The TC program then constructs a packet containing the node IDs and their updated embeddings. To minimize the number of packets sent and the overall communication overhead, In-TC Broadcasting batches the embeddings of multiple nodes into a single packet, up to the maximum transmission unit (MTU) size.

For each destination machine, the TC program looks up the network interface or socket associated with that machine in an eBPF map. It then clones the packet using `bpf_clone_redirect()` and redirects the cloned packet to the corresponding interface or socket. The program also updates the destination IP address and port number of the cloned packet to match the receiving machine.

This eBPF-based broadcasting avoids the need to send the same message multiple times from user space, which would incur repeated user-kernel crossings and traversals of the network stack. To handle packet loss, PINCH incorporates a retransmission mechanism with the TC program as described in § III-E.

### D. Caching

The Caching component is responsible for storing the aggregated node embeddings in eBPF maps and reusing them across iterations to avoid redundant communication and computation. It works in conjunction with the In-XDP Aggregation component.

Specifically, when the In-XDP Aggregation component receives a packet containing node embeddings, it first checks whether the embeddings of the same nodes have been aggregated before by querying the eBPF map. If a cache hit occurs, the XDP program directly returns the cached results without performing the aggregation again. If a cache miss occurs, the XDP program performs the aggregation and updates the eBPF map with the new results. The cache entries in the eBPF map are managed in an LRU (Least Recently Used) manner. When the map is full, the least recently used entry is evicted to make room for the new entry. The cache size can be configured based on the available memory and the workload characteristics.

The Caching and Reuse component can effectively reduce the amount of data transmission and the aggregation overhead, especially for iterative GNN training workloads where the same nodes are accessed repeatedly across iterations.

### E. Fault Tolerance

PINCH is designed to be resilient to various types of failures that can occur during distributed GNN training, such as packet loss, packet duplication.

Packet losses are particularly critical in our current implementation, which uses UDP protocols and where PINCH employs `bpf_clone_redirect`. To address packet losses in In-TC Broadcasting, PINCH utilizes a combination of retransmission and sliding window. Each broadcast packet is assigned a unique sequence number. If the sender does not receive an acknowledgement from all receivers within a specified timeout period, it retransmits the packet. Meanwhile, receivers maintain a sliding window of received sequence numbers and discard any duplicates.

For In-XDP Aggregation, the eBPF program is designed to be idempotent, meaning that processing the same packet multiple times does not affect the correctness of the aggregated results. This is achieved by using the eBPF map as a key-value store, where the key is the node ID and the value is the aggregated embedding. If a packet is processed multiple times due to retransmission, the eBPF program simply updates the corresponding value in the map, which has no effect on the final result.

## IV. Limination and Future Work

While PINCH can accelerate distributed GNN training, there are several limitations and directions for future work.

**Flexible Aggregation Functions.** The current implementation of PINCH supports only a limited set of aggregation functions (e.g., sum, mean, max) due to the constraints of the eBPF virtual machine, which does not natively support floating-point operations. While PINCH currently circumvents this limitation by using fixed-point arithmetic, supporting more complex aggregation functions and customizable aggregators is an important direction for future work. This could involve extending the eBPF virtual machine to support floating-point operations or exploring the use of approximate computing techniques to trade off precision for performance.

**Communication Patterns.** The design of PINCH is currently tailored to the message-passing paradigm of GNNs, where the main communication patterns are broadcasting node embeddings and aggregating neighbor information. Extending PINCH to support other types of GNNs and big data applications, such as attention-based GNNs (e.g., GAT) [21], graph convolutional matrix completion (e.g., GCMC) [20], Pagerank, and Mapreduce, may require additional optimizations and modifications to the eBPF programs. Investigating the applicability and performance of PINCH in these variants of GNNs and big data applications is an insightful direction for future research.

**CPU and NUMA Overhead.** PINCH currently does not consider the impact of CPU affinity and NUMA architecture on the performance of distributed GNN training. This is particularly relevant for the In-XDP Aggregation component, where the aggregated results are stored in eBPF maps which reside on a specific CPU core. If the subsequent processing of these results is performed on a different CPU core, especially one located on a different NUMA node, it can lead to increased memory access latency and cause a high number of cache misses. Investigating techniques such as CPU pinning, NUMA-aware scheduling, and cache-aware data placement to mitigate these overheads and improve the efficiency of PINCH is an important direction for future work.

## V. Related Work

**Distributed GNN Training.** Distributed GNN training frameworks have been developed to handle large graph networks efficiently. DistDGL [28] operates on DGL [24], distributing graphs across multiple machines and employing a message-passing system for inter-machine communication. DistGNN [10] focuses on reducing communication costs through efficient graph partitioning. AliGraph [31], by Alibaba, offers a versatile platform for various GNN models and training approaches. ByteGNN [27] optimizes communication and computation through a bytecode-based model. Unlike these, PINCH leverages kernel-level eBPF optimizations, enhancing communication speeds significantly in distributed GNN training.

**eBPF Applications.** Enhanced Berkeley Packet Filter (eBPF) facilitates advanced networking and systems operations. CCP [12] integrates eBPF's JIT capabilities for effective congestion control through real-time data path measurements. BMC [4] uses eBPF for a kernel-level cache, boosting UDP Memcached GET request throughput. Syrup [7] employs eBPF maps for seamless data exchange across system layers, supporting custom scheduling policies. SPRIGHT [15] enhances sidecar proxy performance in serverless architectures via eBPF-driven fast packet forwarding. XRP [29] offloads kernel storage operations like B-tree lookups to eBPF, reducing overhead. SynCord [13] applies eBPF for dynamic, hardware-aware kernel lock strategies, optimizing synchronization. PINCH targets communication bottlenecks in distributed GNN training, using eBPF and kernel hooks to enhance operations not covered by existing eBPF optimizations.

## VI. Conclusion

In this paper, we present PINCH, a system that optimizes distributed GNN training by leveraging eBPF and kernel-level packet processing for efficient broadcasting and aggregation and enhances performance through caching, and fault tolerance. PINCH has the potential of integrating the communication layer with the kernel network stack, potentially improving components like parameter synchronization and model training in distributed GNN systems. Future plans include expanding PINCH to support advanced GNN models and exploring eBPF for other distributed system challenges.

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
