# OpenReview forum: "PINCH: Accelerating Distributed GNN Training through In-Kernel Operation Using eBPF"
_iscaconf.org/ISCA/2024/Workshop/MLArchSys — MLArchSys 2024 OralPoster_

### Official Review · Reviewer_BZzu · 2024-05-21
**Creative application of eBPF to distributed GNN training**

**Confidence:** 4
**Rating:** 7

**Detailed Feedback And Questions For Authors:**

Summary: This paper proposes PINCH, a framework for distributed GNN training which addresses the high communication overheads for neighborhood aggregations across training nodes. PINCH does so by reducing the overheads of the traditional Linux network stack (e.g., context switches, data copies between kernel/userspace) by performing aggregations and broadcasts within the kernel via eBPF and associated hooks (XDP, TC). PINCH also provides an in-XDP cache for aggregations to reduce re-computation.

Overall, the paper presents a creative solution to a real issue (communication overheads in distributed GNN training). My only major concern is its limitation to CPU-only training. Some detailed comments below.

- A discussion of PINCH’s applicability for GPU training would be useful. In a similar vein, would PINCH be applicable for communication libraries (e.g., gloo) that rely on host networking?

- Similarly, more discussion on related work in GNN training (e.g., BGL NSDI’23 for caching) would be good.

- Some discussion and motivation around the impact/overhead introduced by the kernel/userspace overheads would be useful. While this is a workshop paper, a sense of impact (1% or 100% potential improvement) would make the paper stronger.

**Top Reasons To Accept The Paper:**

- Motivates well and addresses a serious bottleneck (network communication overheads) in distributed GNN training.
- Creatively extends the use of low-level traditional network processing tools like eBPF to ML training, while also addressing and discussing important limitations to its application in this domain.
- Well-written, motivates the background/problem well and provides an easy-to-understand view of the system, especially given length limitations.

**Top Reasons To Reject The Paper:**

- Unclear (quantitatively) how impactful PINCH will be at improving overall training throughput or how significant of an overhead PINCH eliminates.
- PINCH seems to be limited to GNN training frameworks that use CPUs for training (or at least perform comms over the host network stack as opposed to NVLink/RDMA/GPUDirect).

---

### Official Review · Reviewer_SnJB · 2024-05-27
**Could be a good paper if an evaluation section with solid results was provided**

**Confidence:** 3
**Rating:** 3

**Detailed Feedback And Questions For Authors:**

This paper focuses on key challenges in distributed GNN training: heavy communication overhead, scalability, and reliability. Although the motivation is solid and the methodologies are clarified well, this paper does not provide any evaluation data. Quantitative comparison is a key aspect that supports the technical merit of a research paper; without that, it is not possible to properly evaluate this paper.

**Top Reasons To Accept The Paper:**

- Tackles important problems in distributed GNN training
- The paper frontend (introduction, background, and methodologies) is well-written

**Top Reasons To Reject The Paper:**

- The paper is incomplete; no evaluation section.

---

### Official Review · Reviewer_D7U5 · 2024-05-28
**PINCH: Adopts Networking Techniques for GNN Distributed Training**

**Confidence:** 3
**Rating:** 4

**Detailed Feedback And Questions For Authors:**

Thanks for submitting the paper. It is solving a key problem in distributed GNN training. However, I feel that there is a lack of details of the overall architecture being proposed. Some minor comments below:

1.	The point about high communication overheads in GNNs is repeated several times in Introduction, background and motivation – seems redundant.
2.	Should expand XDP in the introduction.
3.	Extra period after advanced packet manipulation in Section II.B
4.	Section III- A is not needed as the authors provided an overview right before it - redundant
5.	Issue - Accuracy challenge of quantized embeddings? Could be helpful to discuss the details.
6.	It would be have been helpful to provide some background on how GNN training workloads where the same nodes are accessed repeatedly across iterations. This would help understand why caching is needed.
7.	Does the sliding window timeout incur performance overhead?
8.	Typo in Section IVA : While PINCH can “accelerating” distributed GNN training,

**Top Reasons To Accept The Paper:**

1. The paper attempts to solve a key problem in distributed GNN training systems where there are is communication overhead.
2. The work leverages state-of-the-art networking technology such as eBPF and XDP to solve the problem of multiple copies in packet processing.
3. The paper proposes an architecture for enabling high-performance GNN training.

**Top Reasons To Reject The Paper:**

1. Lack of implementation details and evaluation study. No results of any form - modeled or implemented - were mentioned in the paper.
2. It is unclear if the quantized embeddings lead to accuracy loss of the system - there was no discussion on commentary on it.
3. A lot of space in the paper was spent in discussing other works rather than the key idea of this paper - could have spent some more space detailing the implementation aspects of this work.

---

### Official Review · Reviewer_WGAL · 2024-05-29
**The paper introduces PINCH, a system that leverages eBPF, XDP, and TC to optimize communication in distributed GNN training. The key techniques employed by PINCH include in-kernel broadcasting and aggregation, kernel-bypass with AF XDP, and caching and reuse of aggregated embeddings. The authors aim to reduce communication overhead, improve scalability, and ensure robustness in the training process.**

**Confidence:** 3
**Rating:** 4

**Detailed Feedback And Questions For Authors:**

1-The paper does not include any experimental results. It does not have any quantitative comparison with existing distributed GNN training frameworks.
2-It was not clear to me that either they use mini-batch training or full-batch training or their approach is orthogonal to this.
3-Many distributed training algorithms rely on sampling to enable scalability issues. I am not sure how the framework is scalable for large models without sampling.  Alternatively, the authors may claim their framework supports both of them. Nonetheless, they need to explain and discuss this matter.
4-The paper also needs to talk about the impact of the employed approach on accuracy.
5-I would like to see how the proposed technique handles large sparsity associated with the current GNN.
6-The paper needs to discuss how it handles different GNNs such as GCN, GIN, GAT? Does it do anything with different GNN algorithms?
7-It is important to examine the overhead of the system in depth, including CPU, memory, and network resources. Analysis of different hardware configurations and workload characteristics is necessary to identify potential bottlenecks and optimize resource allocation.
8-I recommend authors provide the source code and more details of implementation in the appendix.
9-The topics of flexible aggregation functions, optimizing for different communication patterns, and mitigating CPU and NUMA overheads need to be discussed more.

**Top Reasons To Accept The Paper:**

1- The use of eBPF and kernel hooks to accelerate distributed GNN training is a novel solution to address the communication bottleneck. As described communication-intensive operations can be offloaded to the kernel space and scalability can be improved through techniques including caching and kernel-bypass.
2-This article explores communication problems involved in distributed GNN training, including scalability and fault tolerance.

**Top Reasons To Reject The Paper:**

1-The paper does not include any experimental results. It does not have any quantitative comparison with existing distributed GNN training frameworks.

---

### Decision · Program_Chairs · 2024-05-30

**Decision:**

Accept (Oral/Poster)

**Comment:**

Congratulations! We are pleased to inform you that your paper has been accepted for presentation at MLArchSys 2024. We look forward to your participation at the workshop. Further details regarding the schedule and format will be provided soon. See you at the workshop!